**PROCEEDINGS A**

fluid mechanics, applied mathematics

Euler equations, finite-time singularity, pressure, swirl, primitive variables, wall-bounded

**Author for correspondence:**
Dwight Barkley
e-mail: D.Barkley@warwick.ac.uk

# A fluid mechanic's analysis of the teacup singularity

Dwight Barkley

Mathematics Institute, University of Warwick, Coventry CV4 7AL, UK

DB, 0000-0003-4317-3705

The mechanism for singularity formation in an inviscid wall-bounded fluid flow is investigated. The incompressible Euler equations are numerically simulated in a cylindrical container. The flow is axisymmetric with the swirl. The simulations reproduce and corroborate aspects of prior studies reporting strong evidence for a finite-time singularity. The analysis here focuses on the interplay between inertia and pressure, rather than on vorticity. The linearity of the pressure Poisson equation is exploited to decompose the pressure field into independent contributions arising from the meridional flow and from the swirl, and enforcing incompressibility and enforcing flow confinement. The key pressure field driving the blowup of velocity gradients is that confining the fluid within the cylinder walls. A model is presented based on a primitive-variables formulation of the Euler equations on the cylinder wall, with closure coming from how pressure is determined from velocity. The model captures key features in the mechanics of the blowup scenario.

## 1. Introduction

In 1926, Einstein published a short paper explaining the meandering of rivers [1]. He famously began the paper by discussing the secondary flow generated in a stirred teacup—the flow now widely known to be responsible for the collection of tea leaves at the centre of a stirred cup of tea. In 2014, Luo & Hou presented detailed numerical evidence of a finite-time singularity at the boundary of a rotating, incompressible and inviscid flow [2,3]. The key to generating this singularity is the teacup effect. The present work is not aimed at proving the existence of a singularity for this flow, nor is it aimed at generating more highly resolved numerical evidence for the singularity than already exists. Rather, I assume that

the flow simulated by Luo and Hou genuinely develops a singularity in finite time. My goal is to understand, from a fluid-mechanics perspective, why.

## 2. Preliminaries

### (a) Problem statement

The flow under investigation is depicted in figure 1. The system is initialized with a pure azimuthal flow (swirl) having a sinusoidal dependence on the axial coordinate $z$. A pressure field is instantaneously generated to provide the radially inward force necessary to keep fluid parcels moving along circular paths. This results in high pressure at the cylinder wall where the circulation is largest ($z = \pm L/4$) and low pressure where there is no azimuthal flow ($z = 0$ and $z = \pm L/2$). Necessarily, then, there is a vertical variation in the pressure at the cylinder wall and this drives a secondary meridional flow. This is the teacup effect—the portion of the fluid just from $z = 0$ to $z = L/4$ corresponds to a cup of tea. (In an actual cup of tea, the variation in swirl with $z$ is due to a boundary layer at the bottom of the cup. Here, we disregard viscous effects even though they play a role in the motion of real tea in a teacup.)

We consider inviscid fluid flow governed by the incompressible Euler equations

$$\partial_t u + u \cdot \nabla u = -\nabla p \tag{2.1a}$$

and

$$\nabla \cdot u = 0, \tag{2.1b}$$

where $u$ is the fluid velocity and $p$ is the pressure divided by the fluid mass density. By common usage, we refer to $p$ simply as pressure. We work in cylindrical coordinates $(r, \theta, z)$. The flow is axisymmetric (independent of $\theta$), but has swirl ($u_\theta \neq 0$ in general). Hence the velocity has components

$$u(r, z, t) = u_r(r, z, t)\, \hat{e}_r + u_\theta(r, z, t)\, \hat{e}_\theta + u_z(r, z, t)\, \hat{e}_z,$$

where $\hat{e}_r, \hat{e}_\theta$ and $\hat{e}_z$ are standard basis vectors for cylindrical coordinates. The vorticity is $\omega = \nabla \times u$ and has corresponding components $\omega_r(r, z, t)$, $\omega_\theta(r, z, t)$ and $\omega_z(r, z, t)$. The flow takes place inside an axially periodic cylinder of period $L = 1/6$ and radius 1. The boundary condition at the cylinder wall is

$$u_r|_{r=1} = 0. \tag{2.1c}$$

The initial condition employed by Luo and Hou, and reproduced here, is a pure swirl

$$u(r, z, t = 0) = 100 r\, e^{-30(1-r^2)^4} \sin\left(\frac{2\pi}{L} z\right) \hat{e}_\theta. \tag{2.2}$$

This initial condition possesses symmetries that are preserved under evolution of (2.1). The most important is centro symmetry about $z = 0$

$$(u_r, u_\theta, u_z)(r, z, t) = (u_r, -u_\theta, -u_z)(r, -z, t).$$

The full set of symmetry planes is $z_j = jL/4$, $j = 0, \pm 1, \pm 2$; $u_r$ is even and $u_z$ is odd about all these planes; $u_\theta$ is odd about planes $z_o, z_{\pm 2}$ and is even about planes $z_{\pm 1}$. The pressure $p$ is even about all four planes.

Extensive analysis of finely resolved numerical simulations of the Euler equations indicates that starting from the above initial condition, the flow evolves to form a singularity on the critical ring, $(r = 1, z = 0)$, at time $T \simeq 0.0035056$ [2–4]. In the present work, simulations are well resolved to time $t = 0.0031$. Details of the simulations are given in appendix D. I rely heavily on the studies of Luo and Hou (hereafter referred to as LH), to know that the flow at $t = 0.0031$ is indicative of the flow all the way to $t = 0.003505$, extremely close to the singularity time. To be clear, the simulations presented here are not aimed at numerically establishing a singularity (LH have already done this), but instead at understanding the physical mechanisms at work, and for this purpose they are adequate.

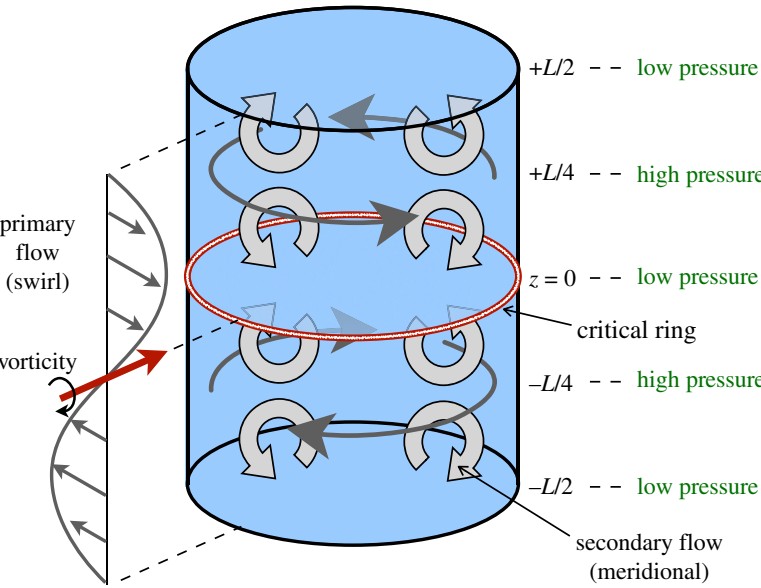

**Figure 1.** Inviscid fluid flow in a cylinder periodic in the axial direction. The primary azimuthal flow (swirl) generates an axial variation in the pressure. This produces a secondary meridional flow that in turn drives azimuthal flow along the cylinder wall towards the critical ring at $z = 0$. The shear of this azimuthal flow generates intense vorticity on the critical ring, ultimately leading to a singularity and a breakdown of the Euler equations. Note that by symmetry, a second critical ring (not indicated) exists at $z = L/2$, which by periodicity is also at $z = -L/2$. The portion of the fluid just from $z = 0$ to $z = L/4$ corresponds to a cup of tea. In the actual configuration studied, the height $L$ is only one-sixth of the radius; see figure 2. (Online version in colour.)

## (b) Mechanics

The pressure is the only stress acting within an inviscid fluid and it is the only means to provide force to, and thereby accelerate, this flow. It is at the heart of the teacup effect and it is therefore natural to investigate its role in the singularity. In general, stress is a tensor field $\tau$ whose divergence gives the net force acting on infinitesimal fluid parcels. Pressure is isotropic and so for inviscid flow $\tau$ has only diagonal components: $\tau_{ij} = -P\delta_{ij}$, where $P$ is the pressure field and $\delta_{ij}$ is the Kronecker delta. Hence $\nabla \cdot \tau = -\nabla P$ is the force per volume acting within the fluid. For incompressible flow, the fluid's mass density $\rho$ is constant and we define $p \equiv P/\rho$, so that $-\nabla p$ sets the fluid acceleration and appears on the right-hand side of the momentum equation (2.1$a$). Subsequently, the symbol $P$ will be used for another quantity and we will refer to $p$ simply as pressure. Pressure gradients with $-\nabla p$ anti-parallel to velocity $u$ are known as adverse pressure gradients and result in flow deceleration (decrease in fluid speed).

The role of pressure in incompressible flow is seen by taking the divergence of (2.1$a$)

$$\partial_t \left( \nabla \cdot u \right) + \nabla \cdot \left( u \cdot \nabla u \right) = -\nabla^2 p. \tag{2.3}$$

This equation governs the evolution of the flow divergence. Given a divergence-free velocity field $u$ satisfying (2.1$b$), in general $\nabla \cdot (u \cdot \nabla u)$ will not be zero, meaning that nonlinearity acting alone does not maintain incompressibility. A pressure field is generated within the fluid (simultaneously everywhere) to accelerate the flow exactly so as to counterbalance this effect of nonlinearity. From (2.3), the relationship between pressure and velocity required to maintain a divergence-free flow is the Poisson equation

$$\nabla^2 p = -\nabla \cdot \left( u \cdot \nabla u \right).$$

This is not the full story, however. The flow of interest is wall-bounded and this puts a condition on the stress field within the fluid. The initial velocity field satisfies (2.1c) and thus has no radial component at the cylinder wall. From the $\hat{e}_r$ component of the momentum equation at the wall, this will be maintained as long as $\partial_r p|_{r=1} = u_\theta^2|_{r=1}$. Thus, the pressure is determined by a Poisson equation together with its boundary condition

$$\nabla^2 p = -\nabla \cdot (u \cdot \nabla u) \equiv S, \quad \partial_r p|_{r=1} = u_\theta^2|_{r=1} \equiv b, \tag{2.4}$$

where these expressions define the source term $S$ and the boundary term $b$. As long as $p$ satisfies (2.4), the flow evolving under (2.1a) will remain incompressible and confined within the cylinder. An important focus of this work will be disentangling the contributions to the stress associated with incompressibility from those associated with flow confinement.

## 3. Basics of the singularity mechanism

Figure 2 presents a quantitative overview of the flow dynamics with visualization of the initial and final states from the numerical simulation. We see the teacup effect: the initial primary flow (pure swirl) results in high pressure on the cylinder wall at $z = \pm L/4$, where the swirl is largest. This, in turn, results in a vertical component of the pressure gradient that produces a secondary flow driving fluid on the cylinder wall toward the midplane $z = 0$ (and by symmetry also toward $z = \pm L/2$). Swirl of opposite signs from above and below the midplane is thus transported towards $z = 0$, resulting in intense radial vorticity $\omega_r = -\partial_z u_\theta$ on the critical ring. This has been described clearly by LH [2–4], who show very strong evidence that the flow continues to develop a singularity in a nearly (but not exactly [5,6]), self-similar way. The final state of the present simulations shown in figure 2 is representative of the flow as it approaches the singularity.

Figure 3a shows an enlargement of the final flow in a very small region around the critical ring. Analysis of this state will be a major focus of the paper. The critical ring is a saddle point of the meridional flow. A local pressure maximum exists on the critical ring (barely visible in figure 2). This secondary local maximum in the pressure field first appears at time $t \simeq 0.002$ and accounts for the stress required to accelerate the flow around the saddle point—bending incoming axial velocity near the cylinder wall to radially inward velocity near the midplane. This pressure field is similar to that reported by LH at $t = 0.003505$, very close to the singularity time $T \simeq 0.0035056$. (See Ref. [3], but note that its figure 17 has a distorted aspect ratio.) LH emphasizes that the pressure maximum on the critical ring means that there is locally an adverse axial pressure gradient decelerating the incoming axial flow on the cylinder wall. This is an important point. However, it does not mean that the pressure maximum inhibits the singularity. On the contrary, a pressure maximum like that in figure 3a can drive a singularity. This fact is central to this work.

To understand the mechanics of this particular situation, we turn to the velocity-gradient dynamics on the critical ring. Differentiating velocity gives the velocity-gradient tensor $\nabla u$ and differentiating the pressure gradient gives the pressure Hessian $\nabla(\nabla p)$. Symmetries dictate that on the critical ring the only non-zero derivatives entering these are

$$W \equiv \partial_z u_z|_c, \quad \Omega \equiv \partial_z u_\theta|_c, \quad V \equiv \partial_r u_r|_c \tag{3.1a}$$

and

$$P \equiv \partial_{zz} p|_c, \quad Q \equiv \partial_{rr} p|_c, \tag{3.1b}$$

where $|_c$ means evaluated on the critical ring. $P$ and $Q$ will be especially important in what follows. I refer to these as *pressure curvatures* since they are the principal curvatures of a graph of the pressure $p(r, z)$. It is to be understood that I always mean 'on the critical ring' when referring to these curvatures. Straightforward differentiation of (2.1a) gives

$$\dot{W} + W^2 = -P, \quad \dot{\Omega} + W\Omega = 0, \quad \dot{V} + V^2 = -Q.$$

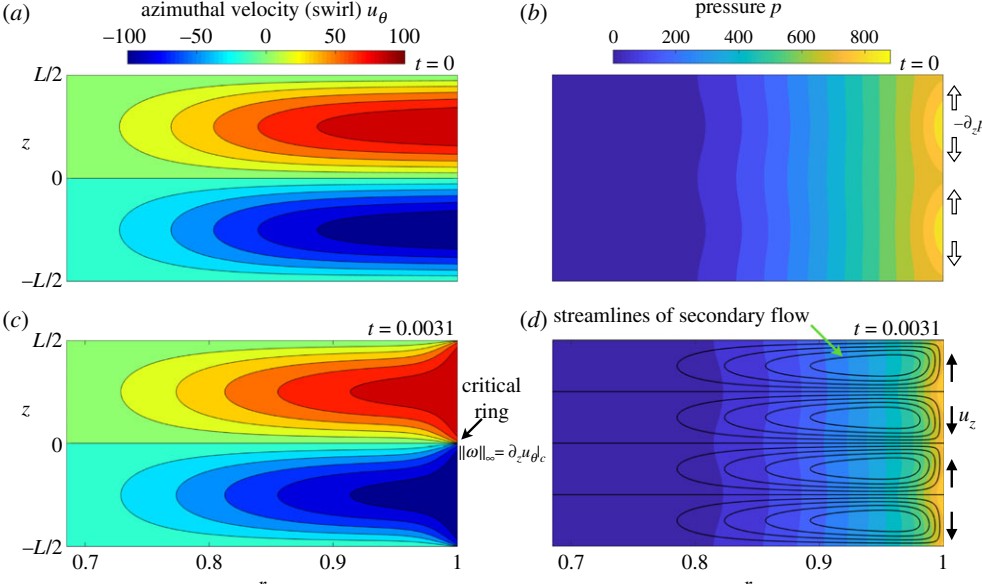

**Figure 2.** Overview of the flow dynamics—similar to the sketch in figure 1 but for the actual flow configuration. (*a*, *b*) Initial azimuthal velocity (swirl) from (2.2) and corresponding pressure field in a meridional plane. The initial swirl is concentrated near the cylinder wall ($r = 1$) and only the outer third of the radius is shown. Axial variation of this primary flow results in an axial pressure gradient (indicated by open arrows) that drives a secondary meridional flow. (*c*, *d*) Azimuthal flow, pressure field and secondary meridional flow at the final simulation time $t = 0.0031$, the standard case analysed in this paper. The meridional flow is shown by contours of the Stokes streamfunction. The surfaces $z = 0$, $z = \pm L/4$, $z = \pm L/2$ and $r = 1$ are flow invariant. Arrows indicate the direction of the meridional flow along the cylinder wall. Advection of $u_\theta$ along the cylinder wall by the secondary flow results in intense radial vorticity $\omega_r = -\partial_z u_\theta$ on the critical ring ($r = 1, z = 0$). (Online version in colour.)

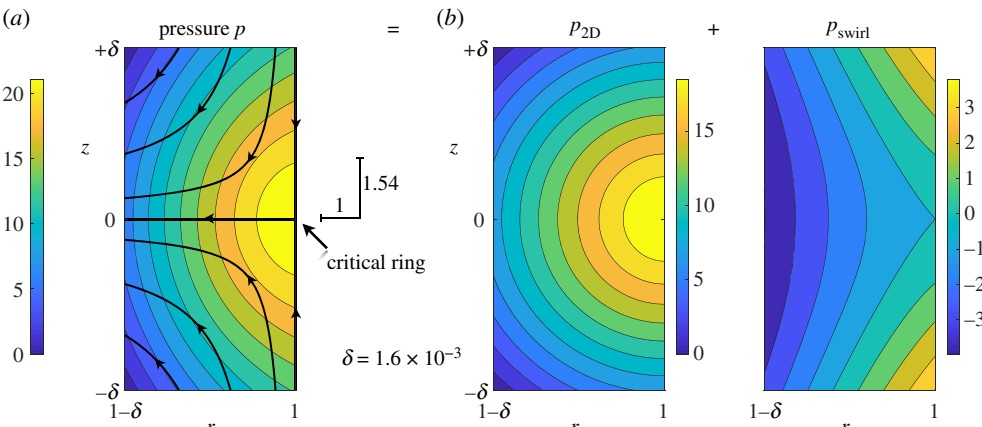

**Figure 3.** (*a*) Enlargement very near the critical ring of the pressure field and meridional-flow streamlines from figure 2*d*. (This region, with $\delta = 1.6 \times 10^{-3}$, will be used in all subsequent plots.) The critical ring is a saddle point for the meridional flow and a local pressure maximum diverts (accelerates) the incoming flow. In particular, the pressure decelerates axial flow $u_z$ on the cylinder wall as it converges towards the critical ring. The length ratio 1.54-to-1 associated with exponent $\gamma = 2.46$ is indicated (see text). (*b*) Pressure $p$ is decomposed into the sum of $p_{2D}$ and $p_{swirl}$, where $p_{2D}$ is determined from the meridional (2D) flow and $p_{swirl}$ from the swirl $u_\theta$. The contours of $p_{2D}$ are nearly circular arcs centred on the critical ring, while $p_{swirl}$ is a hyperbolic point (saddle) with high pressure along the cylinder wall. (Online version in colour.)

By incompressibility on the critical ring: $V + W = 0$. Thus, $V$ can be eliminated, giving the velocity-gradient dynamics

$$\dot{W} + W^2 = -P \quad \text{(axial momentum)}, \tag{3.2a}$$

$$\dot{\Omega} + W\Omega = 0 \quad \text{(vortex stretching)}, \tag{3.2b}$$

and

$$P + Q = -2W^2 \quad \text{(pressure Poisson or mean curvature)}. \tag{3.2c}$$

The meaning associated with each equation is indicated. The equations are exact, and while they are not closed ((3.2c) is insufficient to determine $P$ and $Q$ separately), they are extremely useful in examining what transpires in singularity formation. For this flow, $-\omega_r|_c = \partial_z u_\theta|_c = \Omega$ is the absolute vorticity maximum [2,3], so $\Omega = \|\omega\|_\infty = \partial_z u_\theta|_c$ as indicated in figure 2. Equation (3.2c) is the pressure Poisson equation evaluated on the critical ring, but equally it is the geometrical statement that the sum of principal curvatures is twice the mean curvature, where the mean curvature of $p$ is $-W^2$. (See works by Chae, Constantin & Wu [7–11] for more general treatments of the Euler equations in the velocity-gradient formulation, including several blowup scenarios.)

From figure 3a, we see that the principal pressure curvatures, $P$ and $Q$, are both negative (a pressure maximum occurs on the critical ring), but that they are not equal. The axial curvature is smaller in magnitude than the radial curvature, that is $|P| < |Q|$. This can be seen in the ratio of axial to radial length scales in the pressure contours. This leads us to define $a \geq 0$ by

$$a^2 \equiv \frac{Q}{P}, \tag{3.3}$$

so that $a$ is this ratio of length scales. The case $a > 1$ corresponds to $|P| < |Q|$ and is seen in figure 3a.

To understand the importance of $|P| < |Q|$ to blowup, we proceed as follows. Using (3.3) to eliminate $Q$ from (3.2c) gives $P = -2W^2/(a^2 + 1)$, which can then be used to eliminate $P$ from (3.2a). The velocity-gradient equations (3.2) then become

$$\dot{W} = -\frac{W^2}{\gamma}, \quad \dot{\Omega} = -W\Omega, \tag{3.4}$$

where

$$\gamma \equiv \frac{a^2 + 1}{a^2 - 1} = \frac{Q + P}{Q - P}. \tag{3.5}$$

The case of interest $\infty > a > 1$ corresponds to $1 < \gamma < \infty$.

The solution to equations (3.4) with $\gamma$ constant is simple and captures the essence of the blowup described by these equations. Effectively, we set $\gamma$ to its limiting value, assumed to be finite, as $t \to T$. We are interested in a saddle point flow in the meridional plane, with fluid converging towards the critical ring in the axial direction. We are only ever interested in this situation and always assume $W(t = 0) = W_0 < 0$. The solution to equations (3.4) is then

$$W(t) = -\frac{\gamma}{T - t} \sim (T - t)^{-1}, \quad \Omega(t) = \frac{\Omega_0 T^\gamma}{(T - t)^\gamma} \sim (T - t)^{-\gamma}, \tag{3.6}$$

where $T = -\gamma/W_0 > 0$ is the singularity time and $\Omega_0 = \Omega(0)$. These are the known divergences as $t \to T$ [2,3]. In particular, the vorticity $\Omega = \|\omega\|_\infty$ diverges with exponent $-\gamma$. All other divergences associated with the singularity follow from invariances of the Euler equations and the value of $\gamma$. By treating $\gamma$ as a constant, we obtain the scaling of the blowup as a simple exact solution to (3.4). This will be useful in what follows.

We know from LH that the vorticity diverges with exponent $\gamma \simeq 2.46$, corresponding to $a \simeq 1.54$. The corresponding ratio of length scales is indicated in figure 3a. The contours do not exactly manifest this ratio of scales, in part because contours are a finite distance from the critical ring and in part because the flow is seen at a time a finite distance from the singularity time. From the data at $t = 0.0031$, $\sqrt{Q/P} \simeq 1.62$. (See data in table 1.)

The fundamental point is the following. Incompressibility locks axial contraction and radial expansion together such that it is not the signs of principal pressure curvatures $P$ and $Q$ that are important for singularity formation: it is their inequality. Persistent inequality in the pressure

**Table 1.** Tabulated principal pressure curvatures and other quantities at $t = 0.0031$.

| curvature | value | curvature | value | quantity | value |
|---|---|---|---|---|---|
| $P$ | $-1.9877 \times 10^7$ | $Q$ | $-5.1872 \times 10^7$ | $\partial_r p_a\vert_c$ | $-8.6991 \times 10^3$ |
| $P_{2D}$ | $-2.8556 \times 10^7$ | $Q_{2D}$ | $-4.3192 \times 10^7$ | $\partial_r p_c\vert_c$ | $8.6991 \times 10^3$ |
| $P_{swirl}$ | $8.6797 \times 10^6$ | $Q_{swirl}$ | $-8.6797 \times 10^6$ | $V$ | $5.9895 \times 10^3$ |
| $P_a$ | $9.5265 \times 10^6$ | $Q_a$ | $-9.5178 \times 10^6$ | $W$ | $-5.9895 \times 10^3$ |
| $P_b$ | $-8.4676 \times 10^5$ | $Q_b$ | $5.7998 \times 10^5$ | $\Omega$ | $1.5428 \times 10^5$ |
| $P_c$ | $0$ | $Q_c$ | $2.5808 \times 10^5$ | | |

curvatures on the critical ring can drive the flow to a singularity. Of interest here is flow converging axially toward the critical ring with $|P| < |Q|$, so the axial curvature is smaller than the radial curvature in magnitude. The pressure contours in figure 3a are the signature of this simple mechanism. If the flow evolves such that this situation persists (that is such that $\inf_{t \geq 0} Q/P > 1$), then the solution will blow up. One can deduce from the results of LH that a ratio of pressure curvatures of approximately the same amount as is seen in figure 3a exists as close to the singularity time as they could resolve (figure 17 of Luo & Hou [3]).

## 4. Illustrative cases

Before continuing to a detailed analysis of the pressure field, I consider the velocity-gradient dynamics (3.2) in two limiting cases. These cases will appear again later in the paper, and they are very useful for understanding the interplay between inertia and pressure in the mechanics of the singularity.

Consider simply dropping pressure and the incompressibility constraint from the Euler equations, and also dropping the radial component for the momentum equation. On the cylinder wall, the remaining two components of the momentum equation become Burger's equation and advection of swirl as a passive scalar:

$$\partial_t u_z + u_z \partial_z u_z = 0, \quad \partial_t u_\theta + u_z \partial_z u_\theta = 0.$$

The velocity-gradient dynamics on the critical ring become

$$\dot{W} + W^2 = 0, \quad \dot{\Omega} + W\Omega = 0. \tag{4.1}$$

(Pressure does not appear and (3.2c) is dropped.) Starting with a flow converging towards $z = 0$, $W_0 < 0$, these equations have blowup given by

$$W(t) = -\frac{1}{T - t}, \quad \Omega(t) = \frac{\Omega_0 T}{T - t}, \tag{4.2}$$

where $T = -1/W_0 > 0$ is the singularity time. This is just a special case of equation (3.6) with $\gamma = 1$. In the absence of stresses, there is no deceleration of the fluid parcels converging towards $z = 0$, resulting in the well-known blowup of Burger's equation. This illustrates how inertia, or equivalently the associated advective nonlinearity in Eulerian coordinates, itself can easily lead to a finite-time singularity.

Consider now the case in which principal pressure curvatures are equal at all times: $P = Q$. This would correspond to pressure contours locally forming circular arcs about the critical ring in the meridional plane (similar to what is seen in figure 3b, although those contours are not perfectly circular). With this assumption, equations (3.2) are closed because both pressure curvatures equal

the mean curvature: $P = Q = -W^2$. With this, the velocity-gradient dynamics on the critical ring become

$$\dot{W} + W^2 = W^2, \quad \dot{\Omega} + W\Omega = 0. \tag{4.3}$$

(This case corresponds to $\gamma = \infty$.) The system does not develop a singularity and instead has a solution

$$W(t) = W_0, \quad \Omega(t) = \Omega_0 \exp(-W_0 t). \tag{4.4}$$

Since $W_0 < 0$, the vorticity grows exponentially, but only exponentially in time. This illustrates what is observed to be the normal situation for incompressible inviscid flow—the stress that develops within the flow to maintain incompressibility is such as to accelerate the fluid sufficiently to prevent blowup that would come from inertia acting alone. Algebraically, the term $W^2$ from inertia on the left-hand side of (4.3) is exactly balanced by the term $W^2$ from pressure on the right-hand side.

The case of interest, where $|P| < |Q|$ and hence $1 < \gamma < \infty$, falls between the two extremes just considered. We have a flow configuration evolving under the full incompressible Euler equations, with the pressure stress acting, but such that the axial pressure curvature $P$ is too small to compensate inertia. As a result, fluid parcels converging towards $z = 0$ are not sufficiently decelerated and a singularity occurs.

## 5. Analysis of pressure

In this section, I will analyse in depth the structure of the pressure field near the critical ring and show how it is dictated by specific aspects of the fluid flow. I will then use this information in §6 to gain further insights into the blowup scenario.

### (a) Meridional and swirl pressure fields

We exploit the linearity of the Poisson equation (2.4) to separate pressure into contributions from distinct effects. To begin, the source term for the equation can be decomposed as $S = S_{2D} + S_{swirl}$, where $S_{2D}$ depends only on the meridional (2D) velocity components $(u_r, u_z)$ and $S_{swirl}$ depends only on the swirl velocity $u_\theta$. (See appendix A for details.) The boundary term $b$ in (2.4) also depends only on $u_\theta$. Thus, the pressure $p$ can be written as a linear superposition $p = p_{2D} + p_{swirl}$, where

$$\nabla^2 p_{2D} = S_{2D}, \quad \partial_r p_{2D}\big|_{r=1} = 0 \tag{5.1a}$$

and

$$\nabla^2 p_{swirl} = S_{swirl}, \quad \partial_r p_{swirl}\big|_{r=1} = b. \tag{5.1b}$$

These pressure fields are plotted in figures 3*b* and 3*c*. Contours of $p_{2D}$ are nearly circular arcs indicating approximate rotational symmetry locally about the critical ring within the meridional plane. Contours of $p_{swirl}$ are those of a hyperbolic saddle with the expected high pressure along the cylinder wall where the swirl is largest. Since we will be especially concerned with the axial momentum balance, these fields are plotted along the cylinder wall in figure 4*a*.

Let

$$P = P_{2D} + P_{swirl}, \quad Q = Q_{2D} + Q_{swirl},$$

where $P_{2D} = \partial_z^2 p_{2D}|_c$, $Q_{2D} = \partial_r^2 p_{2D}|_c$, etc., are the principal curvatures of the component pressure fields (see table 1). Since $S_{swirl}|_c = 0$, we have from (5.1b) and (3.2c)

$$P_{2D} + Q_{2D} = -2W^2, \quad P_{swirl} + Q_{swirl} = 0. \tag{5.2}$$

Hence the mean curvature of the pressure field $p$ is contained entirely in the component field $p_{2D}$. (This is obvious since both $p_{2D}$ and the mean curvature $-W^2$ are functions only of the meridional flow, and $p_{swirl}$ is not.) Necessarily then the swirl pressure always has zero mean curvature.

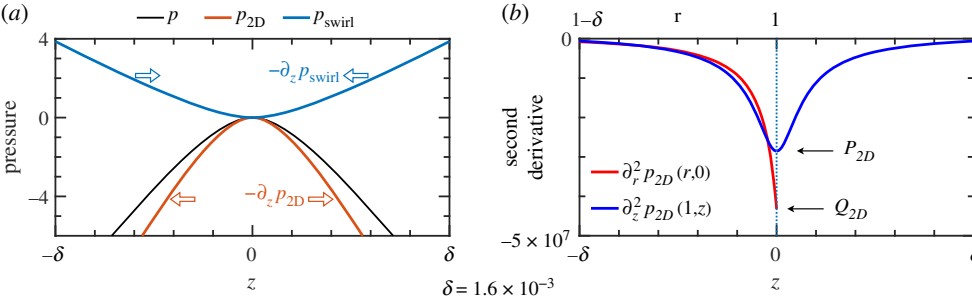

**Figure 4.** (*a*) Pressure components from figure 3 plotted as function of $z$ along the cylinder wall ($r = 1$). Arbitrary constants are such that fields are all zero at $z = 0$. The meridional pressure $p_{2D}$ has negative curvature (local maximum) and hence $-\partial_z p_{2D}$ is an adverse pressure gradient directed outward from the critical ring (open red arrows). The axial curvature of the swirl pressure $p_{swirl}$ is positive and hence $-\partial_z p_{swirl}$ is a favourable pressure gradient directed toward the critical ring (open blue arrows). The full pressure field $p = p_{2D} + p_{swirl}$ has a local maximum, but with less axial curvature than $p_{2D}$, and hence a weaker adverse pressure gradient (not indicated), than $p_{2D}$ near the critical ring. (*b*) Second derivatives of $p_{2D}$ along one-dimensional slices: $\partial_r^2 p_{2D}$ on the midplane and $\partial_z^2 p_{2D}$ on the cylinder wall. The dotted line indicates the critical ring in both cases, where the second derivatives give the pressure curvatures $P_{2D}$ and $Q_{2D}$. (Numerical values for the curvatures are given in table 1.) The near symmetry of $p_{2D}$ in the radial and axial directions does not hold on the critical ring where $|P_{2D}| < |Q_{2D}|$. (Online version in colour.)

The core cause for the inequality in the pressure curvatures, $|P| < |Q|$, is immediately evident. The near symmetry of the meridional pressure maximum implies that $P_{2D} \simeq Q_{2D} \simeq -W^2 < 0$, while for the saddle swirl pressure $P_{swirl} > 0 > Q_{swirl}$. Hence

$$|P| = |P_{2D} + P_{swirl}| < |Q_{2D} + Q_{swirl}| = |Q|. \tag{5.3}$$

Stated simply, the pressure maximum from the meridional flow is flattened by the swirl pressure in the axial direction, but it is steepened by the swirl pressure in the radial direction. This is seen in the visualizations of figure 3 and shown quantitatively along the cylinder wall in figure 4*a*. To fully exploit this insight, more detailed information is required on the meridional and swirl pressure fields.

## (b) Meridional pressure

The pressure field $p_{2D}$ exists within the fluid to counter divergences that would otherwise be generated just by the meridional flow. As the radial gradient of this field is zero at the cylinder wall, (5.1*a*), it does not contribute to fluid confinement. It is determined only by the instantaneous state of the meridional flow. In a region around the critical ring, the meridional velocity field is a saddle that is approximately antisymmetric under interchange of the axial and radial directions. See the streamlines in figure 3*a* where the Stokes streamfunction locally satisfies $\psi(r,z) \simeq \psi(1 - z, 1 - r)$ for $z \geq 0$, and similarly for $z \leq 0$. Although the flow cannot globally respect such a symmetry, very near to the critical ring it does, approximately. Such a saddle flow is to be anticipated [12–14] and the associated approximate rotational symmetry of $p_{2D}$ near a local maximum is not particularly surprising. The source term $S_{2D}$ is quadratic in velocity so an approximately antisymmetric streamfunction implies an approximately symmetric pressure field.

However, it is the principal pressure curvatures on the critical ring that matter for singularity formation. So while the near symmetry of $p_{2D}$ seen in figure 3*b* suggests that $P_{2D} \simeq Q_{2D}$, it is necessary to examine these curvatures quantitatively, in particular to understand in what way they are not exactly equal. Figure 4*b* shows second derivatives of $p_{2D}$ along slices at the midplane, $z = 0$, and at the cylinder wall, $r = 1$. The general agreement between the two curves is

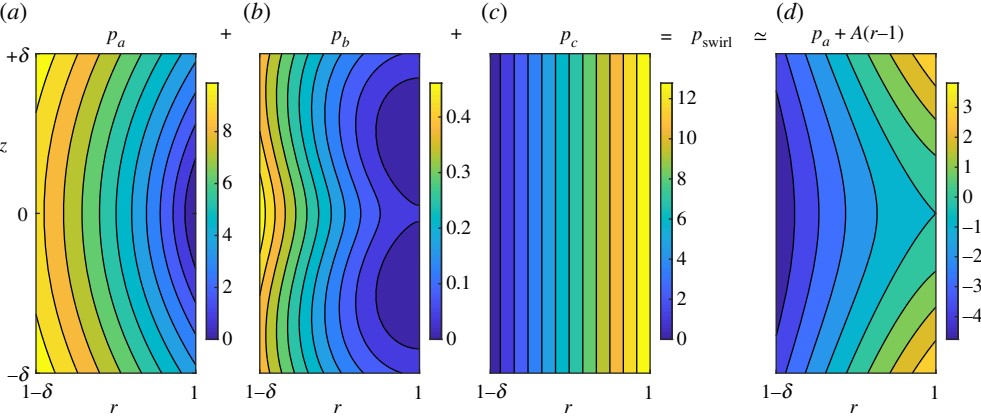

**Figure 5.** Decomposition of the swirl pressure $p_{swirl} = p_a + p_b + p_c$ visualized near the critical ring. (a) The component $p_a$ has a local minimum on the critical ring, but its second derivatives have opposite signs: $P_a > 0 > Q_a$. (b) The range of $p_b$ is smaller than that of either $p_a$ or $p_c$ and its variation along the cylinder wall ($r = 1$) is particularly weak in the region shown. (c) The component $p_c$ does not vary with $z$ by definition and it is nearly a linear function of $r$ in the region shown. (d) The right-most plot is $p_a + A(r - 1)$, where $A = \partial_r p_c|_{r=1} = \langle u_\theta^2 \rangle|_{r=1}$. This field is barely distinguishable from $p_{swirl}$ shown in figure 3. It, and its curvatures $P_a$ and $Q_a$, are determined entirely by the swirl on the cylinder wall. (Online version in colour.)

a manifestation of the near symmetry of $p_{2D}$. However, the curves behave differently approaching the critical ring. Necessarily $\partial_z^2 p_{2D}$ is even about $z = 0$, since $p_{2D}$ is. There is no such constraint on $\partial_r^2 p_{2D}$ at $r = 1$. The important observation is that $Q_{2D} < P_{2D} < 0$, and hence that $|P_{2D}| < |Q_{2D}|$. This means that the pressure curvatures associated with just the meridional flow are unequal with the ordering that promotes, rather than acts against, singularity formation. While this ordering does not seem *a priori* obvious, it appears from figure 4b to be a natural consequence of the conditions at the wall and symmetry plane. We will return to the importance of this ordering in §6c.

## (c) Swirl pressure

The swirl pressure $p_{swirl}$ not only maintains incompressibility of the flow, it also confines the fluid within the cylinder wall. Fully decoupling these two effects is not achievable for flow in a cylinder, but we can mostly separate them via the decomposition $p_{swirl} = p_a + p_b + p_c$, where

$$\nabla^2 p_a = 0, \qquad \partial_r p_a|_{r=1} = \tilde{b}, \tag{5.4a}$$

$$\nabla^2 p_b = \tilde{S}_{swirl}, \qquad \partial_r p_b|_{r=1} = 0 \tag{5.4b}$$

and

$$\nabla^2 p_c = \langle S_{swirl} \rangle, \qquad \partial_r p_c|_{r=1} = \langle b \rangle, \tag{5.4c}$$

where $\langle \cdot \rangle$ denotes axial mean and tilde denotes axial fluctuations. These pressure components are plotted in figures 5 and 6a. We also decompose the pressure curvatures, $P_{swirl} = P_a + P_b + P_c$ and $Q_{swirl} = Q_a + Q_b + Q_c$, with the obvious meanings. (See appendix A for details of this decomposition as well as relationships that hold for the component curvatures.)

The most significant fact from the decomposition is best seen in figure 6a. Near the critical ring, the axial variation of $p_{swirl}$ is given almost exclusively by the component $p_a$. The positive axial curvature of $p_{swirl}$ comes about from the $p_a$ component: $P_{swirl} \simeq P_a > 0$. (The radial curvatures satisfy $Q_{swirl} \simeq Q_a < 0$; see table 1. We return to this shortly.) The stress field associated with the pressure $p_a$ exists throughout the fluid solely to provide force at the wall necessary to confine the flow within the cylinder—the accelerations it generates within the fluid have no effect on the divergence of the flow field. The pressure field $p_a$ is the essence of the teacup effect near the critical ring—axial variation of the swirl at the cylinder wall necessitates a pressure whose radial

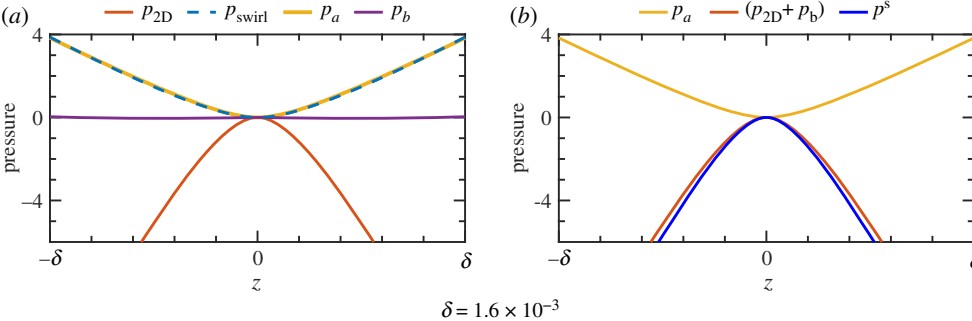

**Figure 6.** Pressure components as function of $z$ along the cylinder wall ($r = 1$). Arbitrary constants are such that fields are all zero at $z = 0$. Since $p_c$ does not vary with $z$, it is zero and not shown. (*a*) Along the cylinder wall near the critical ring $p_a$ and $p_{\text{swirl}}$ are nearly identical. Equivalently, $p_b = p_{\text{swirl}} - p_a$ is very small. The axial curvature of $p_a$ is positive. The axial curvature of $p_b$ cannot be discerned in the plot, but it is negative on the critical ring. (*b*) Justification of the model closure. The two component fields with negative curvatures (adverse pressure gradients), $p_{2D}$ and $p_b$, are summed. (The graph of $p_{2D} + p_b$ is visually indistinguishable from that of $p_{2D}$ since $p_b$ is small.) Also plotted is the field $p_{2D}^s$ generated from the axial velocity of the Euler solution using (6.5). This field from the symmetric approximation has a greater curvature magnitude and generates a larger adverse pressure gradient than the true field $p_{2D} + p_b$. (Online version in colour.)

gradient at the wall confines the fluid and whose axial gradient then forces axial flow toward the critical ring.

The pressure field $p_b$ exists within the fluid to accelerate the flow and suppress divergences that would otherwise arise from spatial variations of $u_\theta$. This component is very weak near the critical ring: the range of values for $p_b$ is small in figure 5 and the curve corresponding to $p_b$ in figure 6*a* is nearly flat. Its curvatures have signs $P_b < 0 < Q_b$ meaning that it acts against singularity formation. However, these curvatures are an order of magnitude smaller than those of $p_a$, so the effect is very weak (see table 1). Further from the critical ring, $p_b$ makes a more substantial contribution to the momentum balance, but this is not important to singularity formation.

The pressure component $p_c$ is easy to interpret physically. It is the axially independent pressure field that would be generated in the pure swirl flow $\sqrt{\langle u_\theta^2 \rangle}(r)\hat{e}_\theta$, whose speed at each $r$ is the axial r.m.s. of $u_\theta$. The radially inward force $-\nabla p_c(r)$ is such as to curve each circular streamline of this r.m.s. swirl flow, both maintaining incompressibility and confining the fluid at the cylinder wall. While this pressure component is significant in the radial momentum balance, it contributes minimally, if at all, to the singularity. By definition $p_c$ does not vary with $z$, so it does not enter the axial momentum balance and $P_c = 0$. While $Q_c$ is not zero, it is the smallest of all component pressure curvatures (see table 1). (I suspect that $Q_c$ does not diverge at the singularity and hence plays no role in the blowup; see appendix A.)

The essential aspect is this: the pressure component $p_a$ is the mechanism by which the swirl on the wall couples to the pressure field. It is the sole pressure component driving singularity formation. It is visually evident in figure 6*a* that $p_a$ is responsible for the positive curvature of $p_{\text{swirl}}$ along the axial direction, and hence the favourable axial pressure gradient. It is less evident comparing figure 5*a* to figure 3*c* that $p_a$ dictates the negative radial curvature of $p_{\text{swirl}}$. This is because $\nabla p_{\text{swirl}}|_c = 0$, while $\nabla p_a|_c \neq 0$. To account for this, in figure 5*d* we show $p_a + A(r - 1)$, where $A = \partial_r p_c|_c = -\partial_r p_a|_c$. The field $A(r - 1)$ is a linear approximation to $p_c$ at $r = 1$, and from the contours in figure 5*c*, $p_c$ is nearly linear in the region shown. Comparing figure 5*d* to figure 3*c* it becomes clear that near the critical ring $p_{\text{swirl}} \simeq p_a + A(r - 1)$. Since the term $A(r - 1)$ is linear, the curvatures of $p_a + A(r - 1)$ are dictated solely by those of $p_a$.

Briefly, the issue here is directly related to the impossibility of separating the interior problem from the boundary condition for the axial mean in a cylinder (5.4*c*). This is only problematic in that $p_a$ does not 'look like' $p_{\text{swirl}}$ because $p_a$ lacks the component of the pressure field responsible for confining the axial mean flow and hence its gradient is non-zero on the critical ring. Conceptually

though, even if one could separate the interior from the boundary for the axial mean, it would not necessarily be desirable to add the resulting field to $p_a$, other than for visual comparisons, because $p_a$ is the fundamental field that alone is responsible for the opposite-signed curvatures driving singularity formation. The axial mean is not important. This decomposition of the swirl pressure is a rich problem that I will not discuss further other than to note that in the Boussinesq system (appendix B and §6) confinement and incompressibility can be fully separated.

## 6. One-dimensional model and closure

I will now use facts learned from the pressure decomposition to gain insight into the mechanics of the blowup scenario. The preceding analysis describes in detail the situation at one time instant, but does not address the persistence of this mechanism as the flow evolves. To do this, I will examine a model based on a primitive-variables formulation of the Euler equations on the cylinder wall, with closure coming from our knowledge of how pressure is determined from velocity.

### (a) Background

There is a rich literature on one-dimensional modelling of singularities in inviscid flow. See [15] for a recent summary. For the cylinder flow, LH propose the model [2,3]

$$\partial_t \omega + u \partial_z \omega = \partial_z \theta, \quad \partial_t \theta + u \partial_z \theta = 0, \tag{6.1}$$

with the identifications $\omega(z) \sim \omega_\theta|_{r=1}$, $\theta(z) \sim u_\theta^2|_{r=1}$ and $u(z) \sim u_z|_{r=1}$. (We abuse notation, by conflicting with usage elsewhere in the paper and by not strictly distinguishing between model quantities and their full-flow counterparts.) Equations (6.1) are closed by determining $u$ from $\omega$ via the Hilbert transform (B 4)

$$\partial_z u = H(\omega). \tag{6.2}$$

The model and closure are natural from a vorticity-formulation viewpoint. The model captures very well features of the teacup flow [3] and exhibits a finite-time singularity [15].

Details arise in the interpretation of the LH model and the model presented below. These are mostly relegated to appendix B. The essential point is that away from the cylinder axis $r = 0$, the axisymmetric Euler equations with swirl have the same structure as the inviscid 2D Boussinesq equations posed on a half-plane (what I shall refer to simply as the Boussinesq system; see appendix B). In particular, because the two systems are equivalent on the wall where the singularity occurs, it is convenient to invoke the structure of the simpler Boussinesq system when considering model closures. In the Boussinesq system, closure models for evolution on the boundary come via the Hilbert transform. In discussing the model below, I will continue to use the language of the axisymmetric Euler equations with swirl, but will invoke the equivalence to the Boussinesq system as needed to close the model.

Of the three variables that appear in the LH model (6.1), two of them, $\omega$ and $u$, are related via the Hilbert transform. One can ask: What about the Hilbert transform of the third variable $\theta$? From (2.4) and (5.4$a$), we have that

$$H(\theta) = H(b) = H(\langle b \rangle + \tilde{b}) = H(\tilde{b}) = H(\partial_r p_a|_{r=1}) = -\partial_z p_a|_{r=1}. \tag{6.3}$$

We have used the linearity of $H$ and $H(\langle b \rangle) = H(\text{const}) = 0$. The final equality is exact, with the understanding that we are invoking the equivalence to the Boussinesq system (see appendix B). Hence the Hilbert transform of $\theta$ is, uniquely, the axial gradient of the pressure field $p_a$ on the boundary. This is the unique physical meaning of $H(\theta)$. Hence, even if one did not set out to study the Euler equations in a primitive-variable formation, one is led to a decomposition of the pressure field just in seeking to understand the meaning of $H(\theta)$. It is important that the variable $\theta$ in the LH model is equivalent to the axial gradient of $p_a$. For any model to capture the correct singularity mechanism, it must capture $p_a$. The LH model does. This helps to explain why the model can so successfully capture the singularity using only variables on the cylinder wall.

## (b) Primitive-variable model

The preceding suggests a different approach to closure: working in a primitive-variable formulation and obtaining pressure by Hilbert transform. In the notation of this section, the Euler equations for the axial and swirl flow on the wall are (exactly)

$$\partial_t u + u \partial_z u = -\partial_z p = -\partial_z p_{2D} - \partial_z p_a - \partial_z p_b \tag{6.4a}$$

and

$$\partial_t \theta + u \partial_z \theta = 0. \tag{6.4b}$$

Recall that $\partial_z p_c = 0$.

These equations can be closed with a single modelling assumption that can be justified from the simulation data. First recall that near the critical ring the contours of the meridional pressure $p_{2D}$ are nearly circular arcs centred on the critical ring (figure 3b). If we assume that $p_{2D}$ is exactly rotationally symmetric about the critical ring in the meridional plane (hence that its contours are exactly circular arcs), then the meridional pressure is expressible just from the source term $S_{2D}$ evaluated on the wall. From this, the associated axial pressure gradient is

$$-\partial_z p_{2D}^s = \frac{2}{z} \int_0^z z' (\partial_z u)^2 \, \mathrm{d}z'. \tag{6.5}$$

See appendix C for details. The superscript $s$ distinguishes this model symmetric meridional pressure from the true meridional pressure. The single modelling approximation we make is to replace the actual adverse pressure gradients $-\partial_z p_{2D} - \partial_z p_b$ in (6.4) by the symmetric pressure gradient $-\partial_z p_{2D}^s$:

$$-\partial_z p_{2D} - \partial_z p_b \rightarrow -\partial_z p_{2D}^s. \tag{6.6}$$

I address the validity of this approximation below.

The favourable pressure gradient $-\partial_z p_a$ is given by the Hilbert transform

$$-\partial_z p_a = H(\theta). \tag{6.7}$$

This is an exact statement with the appropriate interpretation in terms of the Boussinesq system and requires no other assumptions.

Thus, we arrive at the model

$$\partial_t u + u \partial_z u = -\partial_z p_{2D}^s - \partial_z p_a \tag{6.8a}$$

and

$$\partial_t \theta + u \partial_z \theta = 0, \tag{6.8b}$$

where $-\partial_z p_{2D}^s$ is given by expression (6.5) and depends only on the axial flow $u$; $-\partial_z p_a$ is given by expression (6.7) and depends only on $\theta$, the square of the swirl.

The results from simulations of this model are shown in figure 7. The initial condition is

$$u(z, t = 0) = -\frac{z}{1 + z^2}, \quad \theta(z, t = 0) = \frac{1}{2} \frac{z^2}{1 + z^2}, \tag{6.9}$$

where the factor $1/2$ is included in $\theta$ because with it the solution almost immediately exhibits scaling behaviour.

Consider first just the case labelled 'full model', meaning the model as written in (6.8). The dynamics is illustrated in figure 7b with snapshots of the axial velocity $u$, and the swirl velocity $\pm\sqrt{\theta}$. The slopes of these curves at $z = 0$ are $W$ and $\Omega$, the velocity gradients on the critical ring. As the system evolves in time, these gradients steepen from the incoming axial flow.

To establish that the gradients blow up in finite time, we plot $(\Omega/\Omega_0)^{-1/\gamma}$ and $W^{-1}$ as a function of time in figure 7d. Recall the form of the divergence given in equations (3.6) and note that the initial condition has $W_0 = W(0) = -1$. The linearity of these data, together with the common extrapolated zero crossing at $T \simeq 2.250$, is strong evidence that $W$ and $\Omega$ blow up in finite time.

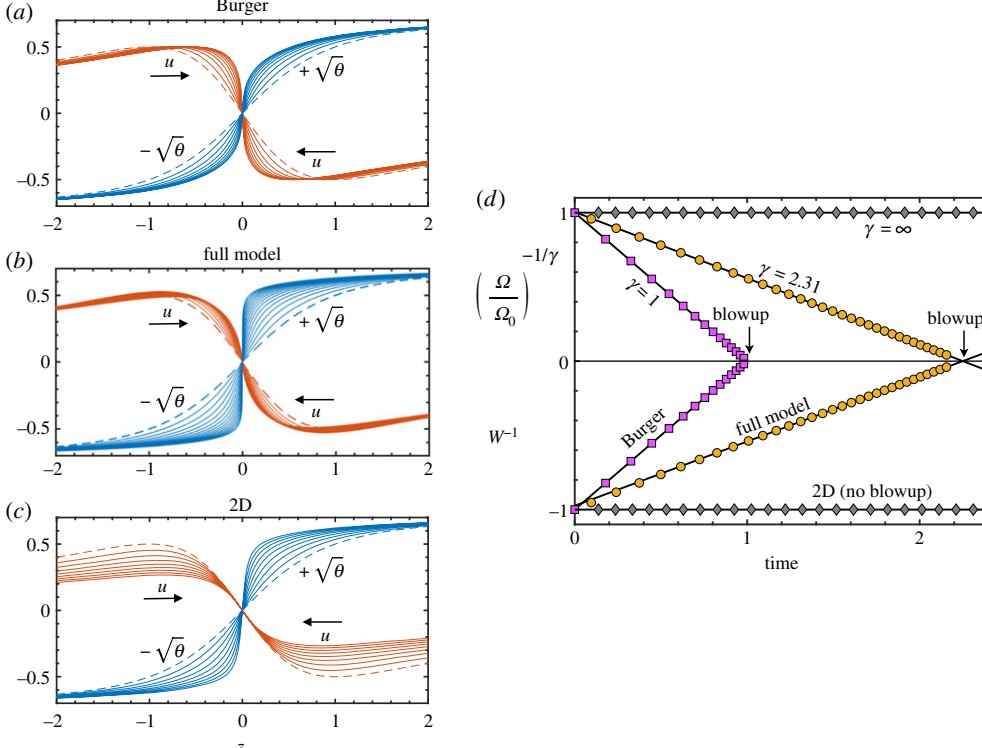

**Figure 7.** Model simulations in three cases: (*a*) Burger's equation (no pressure), (*b*) the full model system (6.8) with both pressure gradients $-\partial_z p_{2D}^s$ and $-\partial_z p_a$, and (*c*) the model with only the meridional (2D) pressure gradient $-\partial_z p_{2D}^s$. Plotted are representative snapshots during the evolution. All start from the same initial condition (dashed curves). The $\theta$ variable is plotted as $\pm\sqrt{\theta}$, since this corresponds to the swirl velocity $u_\theta$ on the cylinder wall. Arrows indicate the flow direction of the axial velocity $u$. The slopes of these curves at $z=0$ are the velocity gradients $W$ and $\Omega$. (*d*) Time evolution of velocity gradients, where gradients are compensated based on the divergences in (3.6). Points are shown at representative times, not every time step. In the Burger case, simulation data agrees well with the known exponent $\gamma=1$ and blowup time $T=1$. For the full model, $\gamma=2.31$ has been estimated from the data (see text). The near linearity of the compensated data with the common extrapolated zero crossing support a blowup at time $T \simeq 2.250$. With meridional pressure only (2D), $W$ is constant, $\gamma=\infty$, and there is no blowup. (Online version in colour.)

The plot of $(\Omega/\Omega_0)^{-1/\gamma}$ requires a value for $\gamma$. This can be estimated from the simulation data in two ways. First, the slope of $W^{-1}$ versus $t$ in figure 7*d* gives an estimate of $1/\gamma$. A least-squares fit of data over the range $1 \le t \le 2$ gives $\gamma \simeq 2.31$. The best-fit line is plotted. A second estimate of $\gamma$ is the value that minimizes the residual error of a least-squares fit of $(\Omega/\Omega_0)^{-1/\gamma}$ versus $t$. Using the fitting range $1 \le t \le 2$, the minimum residual error is obtained for $\gamma \simeq 2.31$, the same value to three digits of accuracy. The plot of $(\Omega/\Omega_0)^{-1/\gamma}$ in figure 7*d* uses $\gamma=2.31$ and the corresponding best-fit line is shown. Note that this exponent is not very different from the value $\simeq 2.46$ obtained by LH for the full Euler simulation.

Before discussing the implications of the model singularity for full Euler flow, I want to return to the two special cases introduced in §4. The model contains both. Dropping all the pressure terms, the model (6.8) reduces to Burger's equation together with the advection of the passive scalar $\theta$. This system has a finite-time singularity with divergences given in (4.2). This singularity is shown in figure 7*a*,*d*. The data have been obtained from a computer simulation that differs from the full model simulation only in the non-evaluation of the pressure terms within the computer code.

The second special case corresponds to the situation with equal axial and radial pressure curvatures on the critical ring: $P = Q$. For the model, $-\partial_z p_{2D}^s$ is obtained under the assumption that the pressure field $p_{2D}$ is exactly rotationally symmetric in the meridional plane about the critical ring (that the contours in figure 3b are exactly circular arcs). This assumption implies that the axial and radial curvatures are equal. Hence dropping only $-\partial_z p_a$ from the model (6.8), but keeping $-\partial_z p_{2D}^s$ gives this special case. The dynamics are shown figure 7c,d. Here, there is the strong deceleration of the axial flow, as seen by the ordering of the $u$ snapshots in figure 7c compared with the other two cases. The gradient of $u$ at $z = 0$ is constant: $W(t) = W_0$, as in (4.4). This system does not blow up. Again the data have been obtained from a computer simulation that differs from the full model simulation only in the non-evaluation of the $p_a$ term within the computer code.

The model captures the interplay between inertia and pressure on the cylinder wall. When no stresses are included, the equations blow up in a Burger's singularity and when only the stress associated with the two-dimensional meridional saddle is included, there is no blowup. In both these cases swirl is advected as a passive scalar, leading to vorticity blowing up in the Burger's case and exponentially growing vorticity in the non-blowup case. Between these two is the case of interest, where the stress from the confinement of swirl on the wall is included. Crucially, here swirl is not a passive scalar: as it is advected towards the critical ring by the axial velocity, it generates increasingly large positive pressure curvature such that the total pressure gradient is insufficient to decelerate incoming flow. Inertia overwhelms pressure gradients and blowup occurs. The model clearly shows how the teacup effect from the wall swirl is able to drive a finite-time singularity.

## (c) Connection to Euler

It remains to relate the model to the full Euler flow. The approximation made in the model closure (6.6) encapsulates the difference between the two, and so I begin with this.

Recall the exact Euler equations on the cylinder wall (6.4), where the axial pressure gradient $-\partial_z p$ separates into contributions $-\partial_z p_{2D}$, $-\partial_z p_b$ and $-\partial_z p_a$. These pressure fields are plotted in figure 6b for the Euler solution at the standard time considered in this paper. The two components with negative curvatures have been combined into $p_{2D} + p_b$. The corresponding adverse pressure gradient $-\partial_z(p_{2D} + p_b)$ produces outward force decelerating the axial flow approaching the critical ring. Also plotted is $p_{2D}^s$, the pressure obtained from (6.5) using the actual Euler flow on the cylinder wall. One sees that $p_{2D}^s \leq p_{2D} + p_b \leq 0$, with equality only on the critical ring. This implies that in the vicinity of the critical ring $|\partial_z p_{2D}^s| \geq |\partial_z p_{2D} + \partial_z p_b|$, meaning that the adverse pressure gradient $-\partial_z p_{2D}^s$ based on the symmetry assumption provides more deceleration to the incoming flow than the actual adverse pressure gradient $-\partial_z(p_{2D} + p_b)$.

This justifies the closure approximation (6.6), where the actual pressure fields acting against singularity formation are replaced by a field that *acts more strongly against singularity formation*. In other words, the closure approximation suggests that the model should be *less liable to blow up* than the full Euler equations. This is important if we want to draw inferences about singularities in the Euler equations from singularity formation in the model. We want to know that we have not, at least not in an obvious way, introduced a singularity mechanism through the model closure.

Another way to view the connection between the model and the Euler equations is via the velocity gradient dynamics on the critical ring. Taking the $z$-derivative of model equations (6.8) and evaluating at $z = 0$ gives

$$\dot{W} = -P_a, \quad \dot{\Omega} + W\Omega = 0 \quad \text{(model)}. \tag{6.10}$$

By construction, the curvature of the symmetric pressure $p_{2D}^s$ exactly balances the inertial nonlinearity on the critical ring, leaving only the pressure curvature $P_a = -\partial_z^2 p_a|_{z=0}$ driving the velocity gradient $W$.

For actual Euler flow we have instead the inequality

$$\dot{W} < -P_a, \quad \dot{\Omega} + W\Omega = 0 \quad \text{(Euler)}. \tag{6.11}$$

This follows from (3.2) under the condition that $Q_{2D} < P_{2D} + 2P_b$. This brings us back to the key observation seen in figure 4b, namely that the meridional pressure curvatures are not exactly equal, $P_{2D} \neq Q_{2D}$. From the data in table 1, $P_{2D}$ and $Q_{2D}$ are sufficiently different that the inequality $Q_{2D} < P_{2D} + 2P_b$ holds ($2P_b$ is an order of magnitude smaller than the difference between $P_{2D}$ and $Q_{2D}$). In fact, the inequality follows from the previous observation that $p_{2D}^s \leq p_{2D} + p_b \leq 0$ with equality only on the critical ring.

The difference between the equality in (6.10) and the inequality in (6.11) quantifies the previous point that the closure approximation appears to be safe, in that it does not (obviously) enhance singularity formation over that of Euler flow. (The singularities occur with $W \to -\infty$.) For simplicity of discussion, throughout this section I have not strictly distinguished between model and full-Euler quantities. Here it is essential to be clear. Equations (6.10) and (6.11) use the same symbols, but apply to different (but closely related) systems: (6.10) holds for solutions of the model equations (6.8), while (6.11) holds for solutions of the full Euler equations. More specifically, (6.10) holds exactly by construction; (6.11) holds by numerical observation of the Euler solution and is presumed to hold up to the singularity time.

Although (6.10) and (6.11) do not establish a rigorous relationship between the model and Euler flow, they nevertheless reduce the mechanism for singularity formation, in both cases, to its most basic form. Within the Boussinesq analogy, $P_a$ is given by the same function of wall swirl in both cases. Using (6.7), we have $P_a \equiv \partial_z^2 p_a|_{z=0} = -H(\partial_z \theta)(0)$. Depending on the case, either $\theta$ comes from the solution of the model (6.8) or else from the swirl on the wall from Euler flow. Referring to (3.4), for either system to blow up, the pressure curvature due to swirl on the cylinder wall must diverge as $W^2$. Specifically, we can relate $W^2/\gamma$ to $P_a$ in (6.10) and (6.11) to give

$$-\frac{H(\partial_z \theta)(0)}{W^2} \leq \frac{1}{\gamma}. \tag{6.12}$$

Either system will blow up if the left-hand side remains bounded above zero by any finite amount.

In principle, (6.12) provides a selection mechanism for the exponent $\gamma$. It selects $\gamma$ sharply in the case of the model and bounds $\gamma$ in the case of Euler flow. It is a global condition relating the swirl everywhere on the wall to the velocity gradient on the critical ring. For the model, one can verify numerically that the axial velocity and swirl evolve together such that $-H(\partial_z \theta)(0)/W^2$ gives the value of $\gamma$. However, this is a triviality given the evidence of a singularity already presented in figure 7. Other than numerical simulations, I have been unable to find any convincing arguments or insights into how a particular value of $\gamma$ is selected. I leave this for future work.

# 7. Conclusion

The potential Euler singularity discovered by Luo & Hou [2,3] has significantly advanced our mathematical understanding of finite-time singularities and it provides a concrete, easily reproducible case to explore computationally. Here I have sought to understand this singularity from a mechanics point of view and from this gain physical insights into why this particular flow configuration permits velocity gradients to blow up in finite time.

The analysis focuses on the interplay between inertia and pressure. A direct connection is established between the singularity mechanism and flow confinement. The pressure field at the heart of the teacup effect is present solely to confine the rotating fluid within the cylinder; it is determined only by the swirl on the cylinder wall and it plays no role in maintaining incompressibility of the flow. This field is responsible for unequal axial and radial pressure curvatures on the critical ring. This inequality of pressure curvatures is precisely the condition

needed for fluid inertia to overwhelm the adverse pressure gradient on the cylinder wall and for velocity gradients to blow up.

To understand how this scenario plays out, a new model has been proposed based on a primitive-variable formulation of the Euler equations. The model describes axial and swirl velocities on the cylinder wall, with closure coming from the dependence of pressure on these velocities. For the swirl, the pressure is known exactly. For the axial velocity, an approximation is made that has a physical meaning and is supported by Euler simulations. This approximation appears to be distinctly different from those used in other models [2,3,13].

The model captures the interplay between inertia and pressure gradients on the cylinder wall and moreover is embedded in a broader class of problems. In one limit, there are no stresses acting and hence no deceleration of axial flow. This leads to an easily understood Burger's singularity, accompanied by vorticity blowup from the transport of swirl as a passive scalar. At the other limit, there is only the stress associated with the acceleration of flow around a two-dimensional saddle point. This leads to substantial deceleration of the axial flow and no blowup. Between these limits is the case that includes both the stress due to the saddle point and the stress generated from confinement of the swirl on the wall (the teacup effect). Swirl is then not a passive scalar: as it is advected towards the critical ring by the axial velocity, it generates an increasingly large positive pressure curvature (and associated favourable pressure gradient) such that the total pressure gradient is insufficient to decelerate incoming flow. Velocity gradients blow up in a singularity.

There is an important connection between this mechanism and other recent popular models for singularity formation [2,3,13]. These models contain two variables, vorticity and square swirl on the cylinder wall. The Hilbert (or similar) transform of the vorticity is used to obtain velocity. The Hilbert transform of the square swirl is, uniquely, the axial gradient of the confining pressure at the core of the mechanism described here.

There are many future directions suggested by this work. Pressure could possibly provide physical insight into the role of the boundary in the rapid growth of vorticity gradients shown by Kiselev & Šverák [12]. The model closure proposed here could be connected to the hyperbolic system studied by Kiselev & Tan [14]. Along these same lines, to impart greater equivalence between axial flow near the cylinder wall and radial flow near the midplane $z = 0$, one could simulate a cylindrical configuration with a no-penetration condition at $z = 0$. The Euler simulations presented here are only for the specific case (initial condition and cylinder aspect ratio) used by Luo and Hou and this leaves open the question of how singularity formation in wall-bounded swirling flows depends on these. One presumes that the scaling exponent $\gamma$ is independent of such factors, as long as blowup occurs, but this is not presently known since the selection mechanism for the exponent $\gamma$ remains open. It would be highly desirable to investigate these issues and to consider other geometries such as swirling flow within a sphere and to understand the role of pressure in other configurations, such as anti-parallel vortices [16]. It seems likely that the blowup observed numerically in the model is self-similar, but this is unknown at present, and currently there is no proof of blowup in the model equations. It should be possible to develop precise theorems along the lines of Chae, Constantin & Wu [7–11] to address the specific pressure fields described here. This could possibly lead to a new line of attack on proof of a singularity in the Euler equations. Finally, and most fundamentally, flow confinement is key to the mechanism described here and hence the question remains open as to whether an Euler solution can exhibit blowup in a configuration without a pressure field originating from flow confinement.

Data accessibility.  The numerical Euler solution at time $t = 0.0031$ and processed data used in plotting the pressure fields are provided as electronic supplementary material.

Competing interests.  I declare that I have no competing interest.

Funding.  This work was partially supported by a grant from the Simons Foundation (grant no. 662985, NG).

Acknowledgements.  I am grateful to Guo Luo for pointing out errors in an earlier manuscript and for providing data with which the present simulations could be validated. I became interested in this problem during the IPAM program on the Mathematics of Turbulence and I thank IPAM for their support.

# Appendix A. Pressure decomposition

Here, we provide details of the pressure decomposition and summarize the relationships that exist between pressure curvatures on the critical ring. In component form, the Euler equations for axisymmetric flow with swirl are

$$\partial_t u_r + \hat{u} \cdot \hat{\nabla} u_r - \frac{u_\theta^2}{r} = -\partial_r p, \tag{A 1a}$$

$$\partial_t u_\theta + \hat{u} \cdot \hat{\nabla} u_\theta + \frac{u_r u_\theta}{r} = 0, \tag{A 1b}$$

and
$$\partial_t u_z + \hat{u} \cdot \hat{\nabla} u_z = -\partial_z p, \tag{A 1c}$$

where $\hat{u} = (u_r, u_z)$ and $\hat{\nabla} = (\partial_r, \partial_z)$.

Taking the divergence of the nonlinear terms gives the source term $S$ on the right-hand side of the pressure Poisson equation

$$S = -\frac{1}{r} \partial_r \left( r \hat{u} \cdot \hat{\nabla} u_r \right) + \frac{1}{r} \partial_r u_\theta^2 - \partial_z \left( \hat{u} \cdot \hat{\nabla} u_z \right).$$

The first and third terms are independent of the swirl velocity $u_\theta$, while the middle term depends only on $u_\theta$. This leads us to define

$$S_{2D} = -\frac{1}{r} \partial_r \left( r \hat{u} \cdot \hat{\nabla} u_r \right) - \partial_z \left( \hat{u} \cdot \hat{\nabla} u_z \right), \quad S_{swirl} = \frac{1}{r} \partial_r u_\theta^2. \tag{A 2}$$

Thus, the pressure Poisson equation, with boundary condition, is

$$\nabla^2 p = S = S_{2D} + S_{swirl}, \quad \partial_r p|_{r=1} = u_\theta^2|_{r=1} = b.$$

This allows for the pressure to be decomposed as $p = p_{2D} + p_{swirl}$, as given in (5.1).

Then, $S_{swirl}$ and $b$ can be further decomposed into axial mean and fluctuating terms

$$S_{swirl} = \langle S_{swirl} \rangle + \tilde{S}_{swirl}, \quad b = \langle b \rangle + \tilde{b},$$

where $\langle \rangle$ denotes axial mean

$$\langle f \rangle (r) = \frac{1}{L} \int_0^L f(r, z) \, dz.$$

This allows for the swirl pressure to be decomposed as $p_{swirl} = p_a + p_b + p_c$, as given in (5.4).

For the velocity gradient dynamics, we require the pressure Hessian $\nabla(\nabla p)$. The pressure field satisfies $\partial_\theta p = 0$ everywhere. Since $p$ is even in $z$, it also satisfies $\partial_z p|_{z=0} = 0$. Hence at $z = 0$ the pressure Hessian is

$$\nabla(\nabla p)|_{z=0} = \begin{bmatrix} \partial_r^2 p|_{z=0} & 0 & 0 \\ 0 & \frac{1}{r} \partial_r p|_{z=0} & 0 \\ 0 & 0 & \partial_z^2 p|_{z=0} \end{bmatrix}$$

with the ordering of components $r$, $\theta$, $z$. On the critical ring, $\partial_r p|_c = 0$ since $u_\theta|_c = 0$, and the pressure Hessian is

$$\nabla(\nabla p)|_c = \begin{bmatrix} Q & 0 & 0 \\ 0 & 0 & 0 \\ 0 & 0 & P \end{bmatrix}.$$

The Laplacian of $p$ is the trace of the Hessian, so on the critical ring $\nabla^2 p = Q + P$.

From the decomposition, the curvatures for the component fields obey

$$P = P_{2D} + P_{swirl} = P_{2D} + P_a + P_b + P_c \tag{A 3}$$

and

$$Q = Q_{2D} + Q_{swirl} = Q_{2D} + Q_a + Q_b + Q_c. \tag{A 4}$$

There are relationships that hold for the component pressure curvatures on the critical ring. From $\nabla^2 p_{swirl}|_c = S_{swirl}|_c = \frac{1}{r} \partial_r u_\theta^2|_c = 0$, we have immediately $P_{swirl} + Q_{swirl} = 0$ leading to (5.2).

Less trivial relationships hold for the decomposition of $p_{\mathrm{swirl}}$ into $p_a + p_b + p_c$. The reason is that $-\partial_r p_a|_c = \partial_r p_c|_c = \langle b \rangle \neq 0$. Hence these terms appear in the pressure Hessian for $p_a$ and $p_c$. From (5.4a), $\nabla^2 p_a|_c = 0$, giving

$$P_a + \partial_r p_a|_c + Q_a = 0. \tag{A 5}$$

From (5.4b) and (5.4c), $\nabla^2(p_b + p_c)|_c = 0$, giving

$$P_b + Q_b + Q_c + \partial_r p_c|_c = 0, \tag{A 6}$$

where we have used that $P_c = 0$. Note that while the second derivatives of pressure blow up at the singularity, the first derivatives do not. This is because $-\partial_r p_a|_c = \partial_r p_c|_c = \langle b \rangle = \langle u_\theta^2 \rangle|_{r=1}$, and $u_\theta$ does not blow up. Hence, close to the singularity

$$P_a + Q_a \simeq 0 \quad P_b + Q_b + Q_c \simeq 0, \tag{A 7}$$

where approximately zero means here that the sums do not diverge even though the individual terms do. The simulations suggest that $Q_c$ does not blow up at the singularity and can be dropped from (A 7). This is reasonable since $Q_c + \partial_r p_c|_c = \partial_r \langle u_\theta^2 \rangle|_{r=1}$, and so for $Q_c$ to blow up, the gradient of the axial mean must blow up. There is possibly an easy demonstration that this cannot occur.

# Appendix B. Connection to 2D Boussinesq system and Hilbert transform

There is a well-known relationship between axisymmetric flow with swirl and two-dimensional thermal convection in the inviscid Boussinesq approximation; see, in particular, [15,17]. The Euler equations for axisymmetric flow with swirl (A 1) can be recast as

$$\partial_t \hat{u} + \hat{u} \cdot \hat{\nabla}\hat{u} = -\hat{\nabla}p + \frac{(ru_\theta)^2}{r^3}\hat{e}_r \tag{B 1a}$$

and

$$\partial_t(ru_\theta) + \hat{u} \cdot \hat{\nabla}(ru_\theta) = 0, \tag{B 1b}$$

where (B 1a) is a vector equation for the meridional flow $\hat{u} = (u_r, u_z)$ obtained by combining (A 1a) and (A 1c). The centripetal acceleration term has been written in terms of $(ru_\theta)^2$ and moved to the right-hand side. Equation (B 1b) is just a reformulation of (A 1b) into a form that expresses conservation of $ru_\theta$ as it is advected as a passive scalar by the meridional flow. Letting $\theta = (ru_\theta)^2$, the equations take the simple from

$$\partial_t \hat{u} + \hat{u} \cdot \hat{\nabla}\hat{u} = -\hat{\nabla}p + \frac{\theta}{r^3}\hat{e}_r \tag{B 2a}$$

and

$$\partial_t \theta + \hat{u} \cdot \hat{\nabla}\theta = 0. \tag{B 2b}$$

In this form, $\theta$ can be viewed as providing a radial driving to the meridional flow. (It should be emphasized, however, that the $\theta$-term in (B 2a) comes from inertia seen in cylindrical coordinates. This term is not associated with stresses acting within the fluid.)

For the inviscid 2D Boussinesq system, consider two-dimensional flow $u(x,y) = u_x(x,y)\hat{e}_x + u_y(x,y)\hat{e}_y$ in the region $y \geq 0$. In the Boussinesq approximation, one allows for density variations within the fluid due to thermal expansion from temperature variations. Gravity acts on the density field, here pointing in the $-\hat{e}_y$ direction, and the governing equations are

$$\partial_t u + u \cdot \nabla u = -\nabla p - \rho \hat{e}_y \tag{B 3a}$$

and

$$\partial_t \rho + u \cdot \nabla \rho = 0, \tag{B 3b}$$

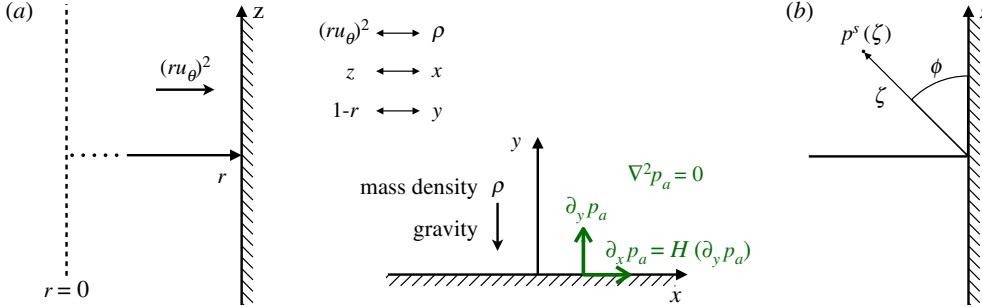

**Figure 8.** (*a*) Correspondence between axisymmetric flow with swirl away from the axis and the inviscid 2D Boussinesq system. Expressions involving $p_a$ illustrate that if $p_a$ is a harmonic function in the upper half-plane, then its tangential derivative is the Hilbert transform of its normal derivative. (*b*) Coordinates for symmetric pressure field $p_{2D}^s$. (Online version in colour.)

where $\rho$ represents the density variation relative to some background density. Equation (B 3*a*) describes momentum balance, while equation (B 3*b*) describes the advection of the density field as a passive scalar. Just as viscosity is zero, thermal diffusivity is zero in this system (both molecular effects are omitted).

The correspondence between the two systems is illustrated in figure 8*a*. Quoting from [17, p. 187], 'we see that the 2D Boussinesq equations are formally identical to the equations for 3D axisymmetric, swirling flows provided that we evaluate all external variable coefficients' … 'at $r = 1$. Thus, away from the axis of symmetry $r = 0$ for swirling flows, we expect the qualitative behaviour of the solutions for the two systems of equations to be identical'.

The advantage of the cylindrical system is that it is straightforward to simulate numerically. The advantage of the Boussinesq system is that it is mathematically simpler. We may consider either system on a periodic or on an infinite domain in the axial, $z$, or horizontal, $x$, direction. The infinite case is the simplest to consider conceptually and it what I mean by the Boussinesq system.

This brings us to the Hilbert transformation. Consider a harmonic function $\phi$ in the upper half-plane. In our case, $\phi$ will be the pressure component $p_a$ associated with the boundary swirl, or boundary density for the Boussinesq system. The Hilbert transform of the normal derivative of $\phi$ along $y = 0$ is the tangential derivative of $\phi$ along $y = 0$. This gives the fundamental relationship between the swirl and axial pressure gradient of $p_a$ on the cylinder wall. Hence, the Hilbert transform appears in §6. A minus sign arises in (6.3) and (6.7) because $y$ is analogous to $1 - r$, so $\partial_y = -\partial_r$. Concretely, the Hilbert transform of a function $f(x)$ is defined as [17, p. 173]

$$H(f)(x) = \frac{1}{\pi} PV \int_{-\infty}^{\infty} \frac{f(x')}{x - x'} \, dx', \tag{B 4}$$

where $PV$ denotes principle value.

## Appendix C. Meridional pressure with symmetry assumption

Assume that in a meridional plane a pressure field $p_{2D}^s$ is exactly rotationally symmetric about the critical ring (that the contours in figure 3*b* are exactly circular arcs). This assumption necessarily requires invoking the Boussinesq analogy because such a symmetry is impossible within a cylinder. As elsewhere, I nevertheless use here the language of the axisymmetric Euler equations with swirl. Let $(\zeta, \phi)$ be polar coordinates centred on the critical ring as shown in figure 8*b*. Then $p_{2D}^s$ is a function only of $\zeta$. We assume $p_{2D}^s$ is determined by a pressure Poisson equation $\nabla^2 p_{2D}^s = S_{2D}^s$, where the source $S_{2D}^s$ must also be rotationally symmetric and hence only a function of $\zeta$. Considering the ray $\phi = 0$ and identifying $\zeta$ with the positive $z$-axis, we set $S_{2D}^s = S_{2D}(r = $

1, z), where $S_{2D}(r = 1, z)$ is the source term for Euler flow (A 2) evaluated on the cylinder wall. Straightforward calculation gives $S_{2D}(r = 1, z) = -2(\partial_z u)^2$, where $u = u_z(r = 1, z)$, from which

$$\nabla^2 p_{2D}^s = \frac{1}{z} \partial_z \left( z \partial_z p_{2D}^s \right) = -2(\partial_z u)^2. \tag{C 1}$$

Integrating this once gives (6.5).

## Appendix D. Numerical simulations

The Euler equations have been simulated in the vorticity-streamfunction formulation as given by equations (2) in [2]. The essential difference between the simulations here and those of Luo & Hou [2,3] is that here a fixed computation grid is used. A Fourier pseudospectral representation is used in $z$ with dealiasing given by Hou & Li [18]. A Chebyshev grid is used in $r$ with no dealiasing. Fourth-order Runge–Kutta time stepping is used with an adaptive time step such that the CFL number is less than 0.2. Exploiting the separation in the Fourier representation, the Poisson problem for the streamfunction is solved directly. Solving similar Poisson problems, pressure fields are computed in a post-processing step.

For all results reported the computation grid has 769 radial points for $r \in [0, 1]$ and 2048 axial points for $z \in [0, L/4]$. At time $t = 0.0031$ simulations produce a vorticity maximum $\|\omega\|_\infty = 1.54276898 \times 10^5$, agreeing to about eight digits of precision with the value $\|\omega\|_\infty = 1.54276901 \times 10^5$ from simulations by Luo and Hou (private communication).

The simulations of the model equations (6.8) are mostly straightforward. The $z$ coordinate is mapped to $x \in (-1, 1)$ via $\lambda \pi z / 2 = \tan(\pi x/2)$, where the parameter $\lambda = 8$ is used to increase the resolution near $z = 0$. The integrals (6.5) and (6.7) are computed by quadrature (taking into account the symmetry of the solution). Derivatives are computed spectrally with 2048 equally spaced grid points in $x$. Fourth-order Runge–Kutta time stepping is used with a time step such that the CFL number is fixed at 0.2.

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
