## [Reviewer comments · Proceedings. Mathematical, Physical, and Engineering Sciences]

Review History

RSPA-2020-0348.R0 (Original submission)

Review form: Referee 1

Is the manuscript an original and important contribution to its field?

Excellent

Is the paper of sufficient general interest?

Excellent

Is the overall quality of the paper suitable?

Excellent

Can the paper be shortened without overall detriment to the main message?

Yes

Do you think some of the material would be more appropriate as an electronic appendix?

No

Do you have any ethical concerns with this paper?

No

Recommendation?

Accept as is

Comments to the Author(s)

See attached file.

Review form: Referee 2

Is the manuscript an original and important contribution to its field?

Good

Is the paper of sufficient general interest?

Good

Is the overall quality of the paper suitable?

Excellent

Can the paper be shortened without overall detriment to the main message?

Yes

Do you think some of the material would be more appropriate as an electronic appendix?

No

Do you have any ethical concerns with this paper?

No

Recommendation?

Accept with minor revision (please list in comments)

Comments to the Author(s)

Report on "A fluid mechanic's analysis of the teacup singularity" by Dwight Barkley.

The present work analyzes the apparent singularity formation in an axis-symmetric flow. The simplicity of the set up allows a deep analysis of the singularity formation and this is work achieves that from a physics point of view. The manuscript is also very clearly written and so I am mostly in favor of publication of this article. I have only some minor comments the authors should address before I recommend publication.

1) The author keeps the same initial conditions as in Luo G & Hou TY 2014 which are not necessarily optimal for the singularity. One can hope that the exact form of the initial conditions might not play an important role close to the singularity if it wasn't for the pressure equation that is non-local. Has the author considered deviations from the Luo & Hou initial conditions? On the same note would a different aspect ratio affect the relation between P & Q and affect the singularity? I am not suggesting any further simulations or analysis but only some discussion.

3) A reminder of the Hilbert transform would help this work to be more complete.

Decision letter (RSPA-2020-0348.R0)

02-Jul-2020

Dear Dr Barkley,

On behalf of the Editor, I am pleased to inform you that your Manuscript RSPA-2020-0348 entitled "A fluid mechanic's analysis of the teacup singularity" has been accepted for publication subject to minor revisions in Proceedings A. Please find the referees' comments below.

The reviewer(s) have recommended publication, but also suggest some minor revisions to your manuscript. Therefore, I invite you to respond to the reviewer(s)' comments and revise your manuscript. Please note that we have a strict upper limit of 28 pages for each paper. Please endeavour to incorporate any revisions while keeping the paper within journal limits. Please note that page charges are made on all papers longer than 20 pages. If you cannot pay these charges you must reduce your paper to 20 pages before submitting your revision. Your paper has been ESTIMATED to be 21 pages. We cannot proceed with typesetting your paper without your agreement to meet page charges in full should the paper exceed 20 pages when typeset. If you have any questions, please do get in touch.

It is a condition of publication that you submit the revised version of your manuscript within 7 days. If you do not think you will be able to meet this date please let me know in advance of the due date.

To revise your manuscript, log into <https://mc.manuscriptcentral.com/prsa> and enter your Author Centre, where you will find your manuscript title listed under "Manuscripts with Decisions." Under "Actions," click on "Create a Revision." Your manuscript number has been appended to denote a revision.

You will be unable to make your revisions on the originally submitted version of the manuscript. Instead, revise your manuscript and upload a new version through your Author Centre.

IMPORTANT: Your original files are available to you when you upload your revised manuscript. Please delete any redundant files before completing the submission process.

In addition to addressing all of the reviewers' and editor's comments, your revised manuscript **MUST** contain the following sections before the reference list (for any heading that does not apply to your work, please include a comment to this effect):

- Acknowledgements
- Funding statement

See <https://royalsociety.org/journals/authors/author-guidelines/> for further details.

When uploading your revised files, please make sure that you include the following as we cannot proceed without these:

- 1) A text file of the manuscript (doc, txt, rtf or tex), including the references, tables (including captions) and figure captions. Please remove any tracked changes from the text before submission. PDF files are not an accepted format for the "Main Document".
- 2) A separate electronic file of each figure (tif, eps or print-quality pdf preferred). The format should be produced directly from original creation package, or original software format.
- 3) Electronic Supplementary Material (ESM): all supplementary materials accompanying an accepted article will be treated as in their final form. Note that the Royal Society will not edit or

typeset supplementary material and it will be hosted as provided. Please ensure that the supplementary material includes the paper details where possible (authors, article title, journal name). Supplementary files will be published alongside the paper on the journal website and posted on the online figshare repository (<https://figshare.com>). The heading and legend provided for each supplementary file during the submission process will be used to create the figshare page, so please ensure these are accurate and informative so that your files can be found in searches. Files on figshare will be made available approximately one week before the accompanying article so that the supplementary material can be attributed a unique DOI. Alternatively you may upload a zip folder containing all source files for your manuscript as described above with a PDF as your "Main Document". This should be the full paper as it appears when compiled from the individual files supplied in the zip folder.

Article Funder

Please ensure you fill in the Article Funder question on page 2 to ensure the correct data is collected for FundRef (<http://www.crossref.org/fundref/>).

Media summary

Please ensure you include a short non-technical summary (up to 100 words) of the key findings/importance of your paper. This will be used for to promote your work and marketing purposes (e.g. press releases). The summary should be prepared using the following guidelines:

*Write simple English: this is intended for the general public. Please explain any essential technical terms in a short and simple manner.

*Describe (a) the study (b) its key findings and (c) its implications.

*State why this work is newsworthy, be concise and do not overstate (true 'breakthroughs' are a rarity).

*Ensure that you include valid contact details for the lead author (institutional address, email address, telephone number).

Cover images

We welcome submissions of images for possible use on the cover of Proceedings A. Images should be square in dimension and please ensure that you obtain all relevant copyright permissions before submitting the image to us. If you would like to submit an image for consideration please send your image to proceedingsa@royalsociety.org

Once again, thank you for submitting your manuscript to Proceedings A and I look forward to receiving your revision. If you have any questions at all, please do not hesitate to get in touch.

Best wishes

Raminder Shergill

proceedingsa@royalsociety.org

Proceedings A

on behalf of

Professor Jonathan Mestel

Board Member

Proceedings A

Reviewer(s)' Comments to Author:

Referee: 1

Comments to the Author(s)

See attached file.

Referee: 2

Comments to the Author(s)

Report on "A fluid mechanic's analysis of the teacup singularity" by Dwight Barkley.

The present work analyzes the apparent singularity formation in an axis-symmetric flow. The simplicity of the set up allows a deep analysis of the singularity formation and this is work achieves that from a physics point of view. The manuscript is also very clearly written and so I am mostly in favor of publication of this article. I have only some minor comments the authors should address before I recommend publication.

1) The author keeps the same initial conditions as in Luo G & Hou TY 2014 which are not necessarily optimal for the singularity. One can hope that the exact form of the initial conditions might not play an important role close to the singularity if it wasn't for the pressure equation that is non-local. Has the author considered deviations from the Luo & Hou initial conditions? On the same note would a different aspect ratio affect the relation between P & Q and affect the singularity? I am not suggesting any further simulations or analysis but only some discussion.

3) A reminder of the Hilbert transform would help this work to be more complete.

Author's Response to Decision Letter for (RSPA-2020-0348.R0)

See Appendix A.

Decision letter (RSPA-2020-0348.R1)

16-Jul-2020

Dear Dr Barkley

I am pleased to inform you that your manuscript entitled "A fluid mechanic's analysis of the teacup singularity" has been accepted in its final form for publication in Proceedings A.

Our Production Office will be in contact with you in due course. You can expect to receive a proof of your article soon. Please contact the office to let us know if you are likely to be away from e-mail in the near future. If you do not notify us and comments are not received within 5 days of sending the proof, we may publish the paper as it stands.

Open access

You are invited to opt for open access, our author pays publishing model. Payment of open access fees will enable your article to be made freely available via the Royal Society website as soon as it is ready for publication. For more information about open access please visit http://royalsocietypublishing.org/site/authors/open_access.xhtml. The open access fee for this journal is £1700/\$2380/€2040 per article. VAT will be charged where applicable.

Note that if you have opted for open access then payment will be required before the article is published – payment instructions will follow shortly.

If you wish to opt for open access then please inform the editorial office (proceedingsa@royalsociety.org) as soon as possible.

Your article has been estimated as being 21 pages long. Our Production Office will inform you of the exact length at the proof stage.

Proceedings A levies charges for articles which exceed 20 printed pages. (based upon approximately 540 words or 2 figures per page). Articles exceeding this limit will incur page charges of £150 per page or part page, plus VAT (where applicable).

Under the terms of our licence to publish you may post the author generated postprint (ie. your accepted version not the final typeset version) of your manuscript at any time and this can be made freely available. Postprints can be deposited on a personal or institutional website, or a recognised server/repository. Please note however, that the reporting of postprints is subject to a media embargo, and that the status the manuscript should be made clear. Upon publication of the definitive version on the publisher's site, full details and a link should be added.

You can cite the article in advance of publication using its DOI. The DOI will take the form: 10.1098/rspa.XXXX.YYYY, where XXXX and YYYY are the last 8 digits of your manuscript number (eg. if your manuscript number is RSPA-2017-1234 the DOI would be 10.1098/rspa.2017.1234).

For tips on promoting your accepted paper see our blog post:
<https://blogs.royalsociety.org/publishing/promoting-your-latest-paper-and-tracking-your-results/>

On behalf of the Editor of Proceedings A, we look forward to your continued contributions to the Journal.

Sincerely,
Raminder Shergill
proceedingsa@royalsociety.org

Appendix A

7 July 2020

I thank the referees for their careful reading of the manuscript and their thoughtful comments.

Response to Referee 1:

The referee asks for my views of the significance of the singularity in light of the fact that the driving mechanism is associated with flow confinement. I am of two minds about the significance of the singularity. On the one hand, it provides the only currently known explicit, reproducible example of a singularity in the 3D incompressible Euler equations. Hence, I now state in the paper (modified slightly from the original submission):

”The potential Euler singularity discovered by Luo and Hou has significantly advanced our mathematical understanding of finite-time singularities and it provides a concrete, easily reproducible case to explore computationally.”

On the other hand, since the driving mechanism is associated with flow confinement, this leaves open the question of singularities in the absence of confinement. Hence, I now state in the paper (modified slightly from the original submission):

”Finally, and most fundamentally, flow confinement is key to the mechanism described here and hence the question remains open as to whether an Euler solution can exhibit blowup in a configuration without a pressure field originating from flow confinement.”

These opinions are tightly connected with the known facts. I do not wish to express any stronger opinions or speculation on this matter in the present publication.

I am grateful to the referee for pointing out the two typos: page 7 line 50 and page 20 line 55. These have been corrected.

Response to Referee 2:

The referee is correct that only one initial condition and one cylinder aspect ratio have been considered. The referee also recognises the work involved in investigating this issue. In fact, the simulations that I have conducted would be inadequate to address fully this issue since they are unable to resolve the flow sufficient close to the singularity time. As suggested by the referee, I have added the following discussion of this point to the paper:

Professor Dwight Barkley, FIMA
Mathematics Institute
The University of Warwick
Coventry CV4 7AL, UK
Telephone: +44 (0) 24 7652 4765
Email: D.Barkley@warwick.ac.uk

"The Euler simulations presented here are only for the specific case (initial condition and cylinder aspect ratio) used by Luo and Hou and this leaves open the question of how singularity formation in wall-bounded swirling flows depends on these. One presumes that the scaling exponent γ is independent of such factors, as long as blowup occurs, but this too is not presently known since the selection mechanism for the exponent γ remains open. It would be highly desirable to investigate these issues and to consider other geometries such as swirling flow within a sphere and to understand the role of pressure in other configurations, such as anti-parallel vortices."

This referee also suggests including the definition of the Hilbert transform. This has been added as equation (7.11) and is referenced in section 6a after the Hilbert transform is first introduced.

Other than these things only minor corrections and some minor wording changes have been made to the manuscript.

Sincerely,
